# Bounded Loss Robustness: Enhancing the MAE Loss for Large-Scale Noisy Data Learning

## Abstract

Large annotated datasets inevitably contain noisy labels, which poses a major challenge for training deep neural networks as they easily fit the labels. Noise-robust loss functions have emerged as a notable strategy to counteract this issue, with symmetric losses, a subset of the bounded losses, displaying significant noise robustness. Yet, the class of symmetric loss functions might be too restrictive, with functions such as the Mean Absolute Error (MAE) being susceptible to underfitting. Through a quantitative approach, this paper explores the learning behavior of bounded loss functions, particularly the limited overlap between the network output at initialization and non-zero derivative regions of the loss function. We introduce a novel method, "logit bias", which adds a real number, denoted as $\epsilon$, to the logit at the correct class position. This method addresses underfitting by restoring the overlap, enabling MAE to learn, even on datasets like WebVision, consisting of over a million images from 1000 classes. Extensive numerical experiments show that MAE, in combination with our proposed method, can compete with state-of-the-art noise robust loss functions. Remarkably, our method relies on a single parameter, $\epsilon$, which is determined by the number of classes, resulting in a method that uses zero dataset or noise-dependent hyperparameters.

## 1 Introduction

Supervised deep learning depends on high-quality labeled data for effective pattern recognition and accurate predictions (Goodfellow et al., 2016). In real-world datasets, however, there is often label noise - erroneous or unclear labels due to human error or incomplete annotations (Liang et al., 2022). Such noise can drastically impair the effectiveness of deep learning models, which often operate under the assumption of pristine labels (Song et al., 2022). Therefore, it is important to develop robust deep-learning algorithms that can efficiently learn from noisy datasets.

One effective approach to navigate label noise lies in employing noise-robust loss functions. These loss functions, notable for their model-agnostic nature, seamlessly integrate with any deep learning paradigm. The existing literature highlights their ability to improve the robustness and generalization ability of deep learning models under noisy conditions (Ghosh et al., 2017; Zhang & Sabuncu, 2018; Wang et al., 2019; Amid et al., 2019; Ma et al., 2020; Zhou et al., 2021; Englesson & Azizpour, 2021).

A majority of these loss functions are bounded to prevent the learning of mislabeled examples. From a theoretical point of view, bounded losses have a higher robustness to noise if they belong to the class of symmetric losses (Ghosh et al., 2017). Nonetheless, it has been suggested that such symmetry could be overly constraining (Zhou et al., 2021), with functions like the Mean Absolute Error (MAE) leaning towards underfitting. Reflecting this, many contemporary loss functions do not satisfy this symmetry condition (Zhou et al., 2021; Englesson & Azizpour, 2021).

In this paper, we quantitatively explore how the vanishing derivatives of bounded loss functions impact their learning behavior. According to our findings, the cause of underfitting is the limited overlap between the output values of an initialized network and the region where the derivative of a particular bounded loss function is nonzero. To counteract this, we add a real number, $\epsilon$, to the logit corresponding to the correct class label. This subtle adjustment reinstates the overlap between network outputs and the region of sufficiently large derivatives of the loss, enabling MAE to surpass the Cross Entropy loss on datasets like Cifar-100, even in the absence of label noise. Impressively,

this approach requires only a single parameter $\epsilon$, which is determined by the number of classes, providing a nearly parameter-free strategy.

Our proposal is intended as a first step towards a universal framework that is capable of noise robust learning across varied class numbers without the need for hyperparameter fine-tuning. This is underscored by our observation that numerous state-of-the-art loss functions, while excelling in benchmarks, heavily rely on intricate hyperparameter adjustments, raising questions about their broader applicability.

In summary, our paper: (i) quantitatively describes how the initial learning phase of a newly initialized network is contingent upon the dataset's class count. (ii) Explores the limitations of bounded losses in multi-class datasets and introduces the "logit bias" technique, enabling MAE to succeed even on challenging datasets like WebVision. (iii) Using numerical experiments, shows that none of the proposed loss functions that are noise resistant on the Cifar-10 dataset are capable of learning the WebVision dataset. In contrast, MAE coupled with the logit bias consistently delivers competitive or even superior results across benchmarks like Fashion-MNIST, Cifar-10, Cifar-100, and WebVision, sans hyperparameters, thus being a first step towards a class count independent framework.

All code for reproducing the data and creating the figures in this paper is open source and available under Ref. Author (s).

## 2 RELATED WORK

Label noise in training data is a pervasive challenge that has attracted much attention in recent years (Liang et al., 2022; Song et al., 2022). One strategy for addressing it is data cleaning, aiming to filter out mislabeled samples from the training dataset. To identify noisy instances, Ref. Xiao et al. (2015) employs a probabilistic model to capture the relationship between images, labels, and noise. Other approaches utilize an auxiliary neural network, trained on curated data, to clean the main dataset (Veit et al., 2017; Lee et al., 2018). Yet, an overzealous curation can sometimes be counterproductive, as eliminating too many samples might degrade model performance (Khetan et al., 2017), compared to retaining some corrupted instances.

Another approach is estimating the noise transition matrix, which depicts the likelihood of mislabeling across classes. This matrix can be incorporated directly into the loss function (Han et al., 2018) or inferred throughout training (Goldberger & Ben-Reuven, 2017; Sukhbaatar & Fergus, 2014), mitigating the consequences of mislabeling. A variation of this strategy involves dynamically adjusting the weights of samples during training. For example, Ref.Reed et al. (2014) adjusts labels based on the network's current predictions, and Ref. Ren et al. (2018) evaluates label trustworthiness based on the gradient induced by a given example. Furthermore, Ref. Thulasidasan et al. (2019) prompts the network to predict the likelihood of the example being correct, enabling fine-tuned loss adjustments.

Another avenue entails constraining the minimum training loss, emphasizing that an optimal scenario that avoids learning from incorrect labels will incur a finite loss (Toner & Storkey, 2023). Self-supervised methods that iteratively adjust labels by extracting information directly from data structures — via dynamic label learning (Chen et al., 2020) and contrastive learning (Hendrycks et al., 2019; Zheltonozhskii et al., 2022; Ghosh & Lan, 2021; Xue et al., 2022; Yi et al., 2022) — have also been shown to improve model generalization in noisy conditions.

Opting for a noise-robust loss function can complement and enhance many of the strategies described above. In Ref. Ghosh et al. (2017) it has been shown that symmetric losses, a subset of the bounded losses, are noise-tolerant. The belief that symmetric losses are prone to underfitting (Zhang & Sabuncu, 2018; Zhou et al., 2021) led to the development of alternative loss functions. Ref. Feng et al. (2021) augments the Cross Entropy loss to render it bounded, and Ref. Lyu & Tsang (2019) introduces the *curriculum loss*, acting as a tight upper bound to the 0-1 loss. Meanwhile, Ref. Xu et al. (2019) advocates for a non-bounded loss based on information-theoretic arguments.

Various combinations have emerged to take advantage of the strengths of multiple losses. The generalized Cross Entropy (genCE) (Zhang & Sabuncu, 2018) synergizes MAE with the Cross Entropy loss (CE) using a Box-Cox transformation, in an attempt to combine the fast initial learning performance of CE with the noise robustness of MAE. On the other hand, symmetric Cross Entropy

(symCE) uses a linear combination of these two losses (Wang et al., 2019). Active-Passive Losses employ linear combinations of MAE with a normalized version of either CE or the Focal loss (Ma et al., 2020). Lastly, the Bi-Tempered Logistic loss (biTemp) *tempers* the exponential and logarithmic functions to create a bounded combination of Tempered Softmax with Tempered Logistic loss (Amid et al., 2019).

## 3 THEORETICAL CONSIDERATIONS

Consider a classification task with $K$ classes, an example space defined as $X$, and associate labels $\{1, 2, ...K\}$. We define a loss function, $\mathcal{L}$, as symmetric if

$$\sum_{y=1}^{K} \mathcal{L}(f(x), y) = C \quad \forall x \in X \quad \forall f, \tag{1}$$

where $C \in \mathbb{R}$ is a fixed constant. Symmetric loss functions have been demonstrated to exhibit robustness against uniform and class-conditional noise (Ghosh et al., 2017). The MAE, for instance, serves as a notable example of a symmetric loss. Such losses fall under the category of bounded losses, implying the existence of a constant $C \in \mathbb{R}$ such that $\mathcal{L}(\boldsymbol{a}, \boldsymbol{y}) < C$, regardless of label $\boldsymbol{y}$ and network output $\boldsymbol{a}$.

A heuristic understanding of this theory can be gained by recognizing that wrongly labeled examples tend to deviate significantly from their class in feature space. In a scenario where these outliers can accumulate unbounded loss, the gradient adjusts the decision boundaries to accommodate even the most extreme outliers. In contrast, a bounded loss ensures that the gradient decreases at greater distances from the decision boundary, preventing overfitting these outliers.

The downside of this approach is a potential slowdown in learning speed, as legitimate examples could be misinterpreted as outliers, with no gradient available to adjust the network weights for accurate classification. The symmetric MAE loss embodies this scenario — it is robust to noise (Ghosh et al., 2017), yet underperforms on datasets like Cifar-100 (Zhang & Sabuncu, 2018).

This section focuses on the early learning stage of a generic neural network and attempts to determine why certain noise-robust loss functions underfit on multiclass datasets (Zhang & Sabuncu, 2018). In addition, we want to identify strategies to address this problem. To explore the underlying causes, we imagine a freshly initialized network with random weights $\mathbf{W}$ drawn from a zero-mean distribution. We focus on the logit distribution for each neuron in the final layer, denoted as $p_z(z_i)$.

Owing to inherent symmetry, we obtain $p_z(z_i) = p_z(z_j) \equiv p_z(z)$. The central limit theorem suggests that when the width $N$ of the last hidden layer is infinite, the logit distribution $p_z$ corresponds to a normal distribution with zero mean (Neal, 1994; Lee et al., 2017). As $N$ is typically quite large, $p_z(z_i)$ can indeed be well approximated by a zero-mean normal distribution.

Under conventional training regimes, where the weights are initialized with variances given by the inverse of the layer output size, initial neuron pre-activations have approximately a unit variance [1]. Therefore, we adopt a $\mathcal{N}(0, 1)$ distribution for the logits $z_i$. Though our results primarily consider this distribution, they can be easily generalized for other variances. For an empirical demonstration of the distribution under standard initializations, we refer to the appendix.

When training neural networks using backpropagation, the error associated with each neuron can be expressed as

$$\delta_n = \frac{\partial \mathcal{L}}{\partial z_n} = \sum_{j=1}^{K} \frac{\partial \mathcal{L}}{\partial a_j} a_j (\delta[j - n] - a_n), \tag{2}$$

where $K$ denotes the number of classes and $\delta[j - n]$ the Kronecker delta. Moreover, we assume that the network output $a_i$ at position $i$ is defined as $a_i = \exp(z_i) / \sum_j \exp(z_j)$ where $z_i$ is the logit at output position $i$ meaning that we assume the softmax activation for the final layer.

---

[1]It is worth noting that there have been instances where smaller initializations have been recommended (Yang et al., 2021)

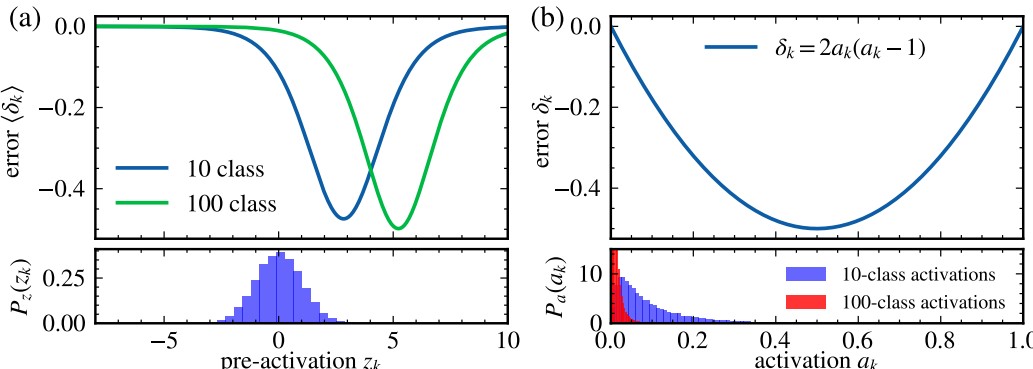

Figure 1: Analysis of the average error for the neuron corresponding to the correct label, determined for the MAE loss. (a) Error as a function of the pre-activation value. As we increase the number of classes from 10 to 100, there is a notable shift in the range where the average error $\langle \partial_{z_k}\mathcal{L} \rangle$ is non-vanishing. However, the logit distribution of a newly initialized network (shown in the histogram at the bottom) remains class-count invariant. This mismatch leads to diminished gradients, stalling the learning due to tiny errors. (b) A perspective shift, showcasing the error as a function of activation, $a_k$. Although the error remains stable across class variations, the distribution of $a_k$ skews towards zero as the number of classes grows.

This error plays a central role in backpropagation. Specifically, it determines the magnitude of the gradient descent updates, expressed as $\nabla_{W_l}\mathcal{L} = \boldsymbol{\delta} \cdot \boldsymbol{a}_{l-1}^T$. An error of zero results in a zero gradient, causing learning to stall.

From our previous discussions, we established that the distribution of $z_i$ corresponds to a normal distribution, $\mathcal{N}(0, 1)$. Consequently, in the initial learning phases the average error of neuron $n$ as a function of the logit $z_n$ of that neuron is given by:

$$\langle \delta_n \rangle(z_n) = \left\langle \sum_{j=1}^{K} \frac{\partial \mathcal{L}}{\partial a_j} \frac{\exp(z_j)}{\sum_i \exp(z_i)} \left( \delta[j-n] - \frac{\exp(z_n)}{\sum_i \exp(z_i)} \right) \right\rangle_{z_i, i \neq n}. \tag{3}$$

To gain insights, we visualize the average error, $\langle \delta_k \rangle$, in the neuron $k$ corresponding to the correct label as a function of its pre-activation. This involves computing the average with respect to $z_{i \neq k}$ and assigning a static value to $z_k$. This perspective offers a snapshot of typical error values in a newly initialized network. Using the MAE as an example, Fig. 1 (a) shows that with 10 classes, the range over which the average error $\langle \delta_k \rangle$ notably deviates from zero overlaps significantly with the initial logit distribution of a network at initialization. However, this overlap diminishes with 100 classes as the error depends on the number of classes.

Fig. 1 (b) describes this dependence using the neuron activation $a_k$ instead of the logit $z_k$. While the error now remains class independent, the distribution $p_a(a_k)$ in the final layer is skewed towards zero for a larger number of classes, i.e. is concentrated in a region with small error. These illustrations underscore the challenge: as the number of classes increases, gradients vanish under the MAE loss. This phenomenon has implications for all bounded loss functions, which have very small gradients for outliers. Importantly, what is deemed an "outlier" varies with the class count. We will later illustrate how this makes learning on the WebVision dataset infeasible for certain noise-robust loss functions optimized for up to 100 classes.

Relevant to this discussion, Ref. Ma et al. (2020) categorizes the active component of a loss function as the contribution that optimizes the output in the correct neuron. This component has been identified as essential to successful fitting. The optimization within the correct neuron turns out to be the linchpin of the fitting process, especially since the errors in the incorrect neurons decrease with increasing number of classes: already for 100 classes, the initial activation of an "incorrect" neuron in the final layer is very close to the optimal value of zero, with an average value of $0.01$, leading to rather small errors in these neurons. This accentuates the need to focus primarily on the error in the "correct" neuron, $\delta_k$.

Table 1: Overview of the examined loss functions, with the neuron associated with the correct label indexed by $k$. The reported number of parameters excludes values that we keep constant for all datasets. Instead, it focuses on values actively adjusted based on the dataset under consideration. Parameter values utilized in our simulations, sourced from respective publications, are detailed in the appendix.

| loss | definition | parameters | output error $\delta_n$ | Ref. |
|------|-----------|-----------|------------------------|------|
| CE | $-\log(a_k)$ | 0 | $a_n - \delta[n-k]$ | - |
| MAE | $2(1 - a_k)$ | 0 | $2a_k(a_n - \delta[n-k])$ | - |
| genCE | $q^{-1}(1 - (a_k)^q)$ | 0 | $(a_k)^q(a_n - \delta[n-k])$ | Zhang & Sabuncu (2018) |
| biTemp | no analytic form | 0 | no analytic form | Amid et al. (2019) |
| NCE | $\frac{\log(a_k)}{\sum_i \log(a_i)}$ | 0 | see appendix | Ma et al. (2020) |
| NF | $\frac{\log\left((1-a_k)^{0.5}a_k\right)}{\sum_i \log\left((1-a_i)^{0.5}a_i\right)}$ | 0 | see appendix | Ma et al. (2020) |
| AGCE | $[(b+1)^q - (b+a_k)^q]/q$ | 2 | $a_k(a_n - \delta[n-k])(b+a_n)^{q-1}$ | Zhou et al. (2021) |
| NF-MAE | $\alpha\,\text{NF} + \beta\,\text{MAE}$ | 1 | see appendix | Ma et al. (2020) |
| NCE-MAE | $\alpha\,\text{NCE} + \beta\,\text{MAE}$ | 1 | see appendix | Ma et al. (2020) |
| NCE-AGCE | $\alpha\,\text{NCE} + \beta\,\text{AGCE}$ | 4 | see appendix | Zhou et al. (2021) |

### 3.1 BOOSTING LEARNING WITH EXAMPLE-DEPENDENT LOGIT BIAS

In Fig. 1, we have illustrated the challenge of using the MAE loss in scenarios with a large number of classes. The root of the challenge is the reduced overlap between the initial logit distribution and the region where the derivative of the loss is substantial. To tackle this, we introduce an example-dependent bias, denoted as $\epsilon$, to the logit, adjusting it from $z_k$ to $z_k + \epsilon$. This adjustment ensures that the learning speed achieved with the MAE loss is preserved irrespective of the class count. As before, $k$ refers to the class specified by the label.

The consequence of this adjustment for a newly initialized network is demonstrated in Fig. 2 (b). Here, the logit distribution, $p_z(z_k)$, shifts from $\mathcal{N}(0, 1)$ to $\mathcal{N}(\epsilon, 1)$, reinstating the overlap with the region where significant gradients can emerge. A noteworthy aspect of the MAE error is its implicit dependency on class count through the relationship $\delta_n = 2(1 - a_k)$. Since the $a_k$ distribution is influenced by the class number, maintaining a consistent $a_k$ distribution would ensure a steady error. To realize this, we aim to solve the following implicit equation for $\epsilon(K)$:

$$\left\langle \frac{\exp(z_k + \epsilon(K))}{\sum_{i=1}^{K} \exp(z_i + \delta[i-k]\epsilon(K))} \right\rangle = C\,, \tag{4}$$

where $C \in [0, 1]$ represents a freely chosen constant which specifies the average activation of $a_k$ after the logit bias is applied. Our choice for $C$ is $C = 0.15$. This particular value ensures a substantial overlap between the network's initial logit distribution and the region where substantial gradients can occur, whilst excluding outliers with respect to the initial distribution.

We can evaluate Eq. (4) numerically for varying class counts. This is achieved by sampling Gaussian numbers, $z_i$, for a given $\epsilon$, and subsequently refining our estimate using techniques such as binary search. Results for the dependence of $\epsilon$ on the class number are presented in the appendix. To clarify, when a logit bias is incorporated into the MAE loss, we label the resulting loss as MAE*.

### 3.2 COMPARISON OF VARIOUS LOSS FUNCTIONS

To address the underfitting described above for datasets with a moderate number of classes, prior work has proposed two main strategies: (i) Implementing an unbounded loss of the form $\mathcal{L} = \alpha\mathcal{L}_{\text{bound}} + \beta\mathcal{L}_{\text{unbound}}$ (Wang et al., 2019), and (ii) Designing bounded losses that decay slowly towards outliers, thus assigning them non-zero gradients (Ma et al., 2020; Zhou et al., 2021; Zhang & Sabuncu, 2018; Amid et al., 2019). In Tab. 1, we show a comprehensive list of noise-robust loss functions that we will use to identify potential problems that arise when learning with these losses. The parameter settings for each of these loss functions, as per their respective publications, are documented in the appendix.

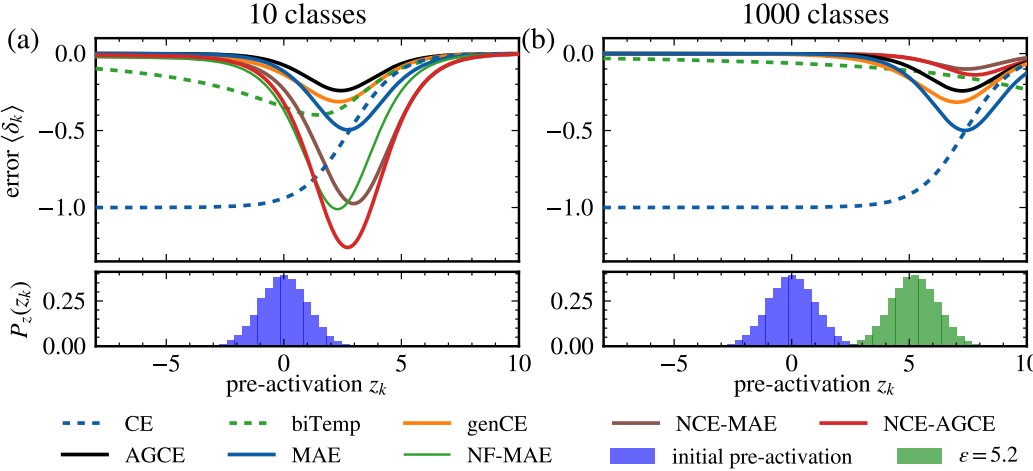

Figure 2: The average error $\langle \delta_k \rangle$ of the final layer's correct neuron $k$, a determinant factor for the magnitude of a gradient descent update, plotted against the pre-activation $z_k$, for a network at initialization. Panel (a) depicts a ten-class learning scenario, showcasing a pronounced overlap between the region where learning is possible (characterized by large negative $\langle \delta_k \rangle$) and the logit distribution from a randomly initialized network (blue histogram). In contrast, in a scenario with 1000 classes shown in panel (b), this overlap is not present for the bounded loss functions. Rather than employing a tailed loss — exemplified by biTemp (delineated by the dashed green line) — our approach involves shifting the $z_k$ distribution into the learning-possible range by adding a bias to the correct neuron's pre-activation (green histogram). We note that the average error of NCE-AGCE is rescaled by a factor of $0.5$ in panel a) to enhance visual clarity.

Fig. 2 shows the average error $\langle \delta_k \rangle$ in the "correct" neuron as a function of its pre-activation across a set of noise-robust loss functions. Panel (a) shows a scenario with 10 classes for a neural network at initialization. The lower section of this panel represents the anticipated logit distribution $\mathcal{N}(0, 1)$ for the neuron $k$. It is obvious that these noise-robust loss functions deviate from the traditional Cross Entropy loss. Specifically, these functions assign gradients of reduced magnitude to examples with higher uncertainty (represented by small $z_k$) than the Cross Entropy loss. Despite this difference, the errors significantly deviate from zero at points of expected initial learning, as the histogram shows.

However, in a 1000-class scenario, the overlap between the initial distribution and the regions where noise-robust losses record substantial errors is almost non-existant. Subsequent sections will show how this characteristic leads to negligible learning rates in the context of the 1000-class WebVision dataset. Already in the case of 100 classes (not shown), the overlap is significantly reduced, slowing down learning on the Cifar-100 dataset.

Instead of using the unbounded Cross Entropy loss, which indiscriminately learns from outliers (indicated by very small $z_k$ values), we suggest an alternative: adjusting the logit distribution to $p_z(z_k) = \mathcal{N}(\epsilon, 1)$, to reestablish the crucial overlap with the error, as visually represented by the green histogram.

## 4 EMPIRICAL RESULTS

In the following, we present empirical evidence highlighting the efficiency of the logit bias in enhancing the learning capability of MAE in multi-class problems, demonstrating competitive or superior performance in the presence of label noise. Additionally, our experimental results closely mirror our theoretical predictions regarding the learning behavior of various loss functions. Datasets employed in this study include the publicly available datasets Cifar-10, Cifar-100 (Krizhevsky et al., 2009), Fashion-MNIST (Xiao et al., 2017), and the web-scraped WebVision dataset (Li et al., 2017). The noise profile in Cifar datasets is inherently minimal, enabling us to precisely adjust the label noise level. In contrast, WebVision naturally offers a more complex, potentially asymmetric noise

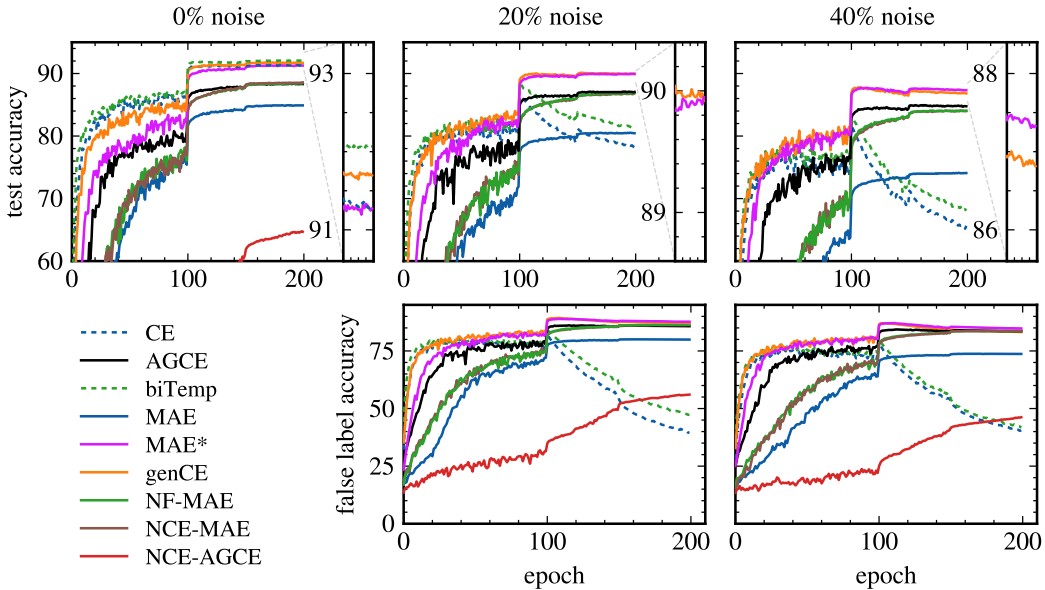

Figure 3: Training results of ResNet-32 on the Cifar-10 dataset using various loss functions under different label noise conditions. Accuracies represent the average across five different network initializations. In the upper panels, test accuracies are plotted against training epochs for label noise levels of 0 % (left), 20% (center), and 40% (right). Insets zoom into the accuracy of top-performing loss functions during the final training epochs. The lower panels display "false label accuracies," which measure the accuracy on images that initially had noisy labels but were corrected for testing.

structure. ResNet architectures (He et al., 2016), tailored in size to the specific dataset, are used for most of our experiments. We create symmetric label noise by assigning random labels to a fraction of the training dataset. Details on the training, together with further results from alternative network architectures, are available in the appendix.

Fig.3 shows the performance of the loss functions outlined in Tab.1 when subjected to the Cifar-10 dataset. The top panels depict test accuracies across training epochs. In the absence of label noise (upper left panel), MAE's learning speed lags behind most other loss functions. Yet, the introduction of a minor logit bias (specifically, $\epsilon = 0.5$) in MAE* noticeably enhances the learning speed, bringing it on par with Cross Entropy and biTemp. In scenarios with label noise, MAE* maintains its noise resilience. In stark contrast, both the Cross Entropy and biTemp losses tend to overfit to the noise, a trait anticipated from our theoretical discussions — especially noting biTemp's pronounced outlier tail in Fig.2. Among the tested functions, only genCE mirrors the robustness exhibited by the modified MAE. Subsequently, the lower panels in Fig. 3 show the accuracy on samples from the training dataset that were mislabeled during training but have now been corrected. Remarkably, MAE* showcases superior resilience to noise, adeptly disregarding inaccurately labeled data and achieving the optimal *false label accuracy*.

The summarized findings, along with results on Fashion-MNIST, are documented in Table 2. This table reports mean accuracies and associated errors (across five seeds) under various levels of label noise. Except for genCE and MAE*, most proposed loss functions either attain diminished accuracies in noise-free scenarios relative to Cross Entropy (CE) or fit the noise to a high degree. The best performing loss functions are emphasized in bold (within error bars) for every combination of network architecture and noise level.

In Fig.4, test accuracies as a function of the training epoch are depicted for the more challenging Cifar-100 dataset. Without additional noise, it becomes evident that only the Cross Entropy, biTemp, AGCE, and MAE* loss functions function well on the dataset — their trajectories notably supersede others. Of this quartet, MAE* emerges as superior, suggesting that even the minimal noise inherent in the Cifar-100 dataset (Northcutt et al., 2021) puts bounded loss functions in a favored position

Table 2: Final test accuracies for training ResNet-32 on the ten-class datasets Cifar-10 and Fashion-MNIST with different loss functions and various amounts of label noise.

| Dataset, Network | Loss | Noise: 0 % | 10 % | 20 % | 40 % | 60 % |
|---|---|---|---|---|---|---|
| | CE | $91.30 \pm 0.33$ | $83.80 \pm 0.36$ | $78.21 \pm 0.42$ | $65.65 \pm 0.23$ | $50.12 \pm 0.66$ |
| | MAE | $84.90 \pm 1.86$ | $82.59 \pm 1.22$ | $80.45 \pm 0.24$ | $74.08 \pm 4.17$ | $59.90 \pm 6.19$ |
| | MAE* $\epsilon = 0.5$ | $91.27 \pm 0.13$ | $90.67 \pm 0.09$ | $\mathbf{89.95} \pm 0.06$ | $\mathbf{87.33} \pm 0.07$ | $81.43 \pm 0.19$ |
| Cifar-10 | AGCE | $88.32 \pm 1.86$ | $87.83 \pm 1.87$ | $87.06 \pm 1.86$ | $84.77 \pm 1.76$ | $\mathbf{82.24} \pm 1.37$ |
| | genCE | $91.70 \pm 0.10$ | $\mathbf{91.12} \pm 0.04$ | $89.92 \pm 0.11$ | $86.81 \pm 0.12$ | $78.88 \pm 0.30$ |
| ResNet-32 | NCE-AGCE | $64.70 \pm 1.51$ | $59.21 \pm 1.13$ | $55.77 \pm 2.44$ | $46.19 \pm 3.56$ | $35.76 \pm 1.85$ |
| | NF-MAE | $88.47 \pm 0.26$ | $87.64 \pm 0.18$ | $86.91 \pm 0.07$ | $84.05 \pm 0.37$ | $75.98 \pm 0.49$ |
| | NCE-MAE | $88.57 \pm 0.09$ | $87.73 \pm 0.14$ | $86.81 \pm 0.08$ | $84.00 \pm 0.19$ | $75.67 \pm 0.57$ |
| | biTemp | $\mathbf{92.07} \pm 0.05$ | $86.84 \pm 0.17$ | $81.33 \pm 0.16$ | $68.06 \pm 0.63$ | $51.58 \pm 0.40$ |
| | CE | $\mathbf{94.90} \pm 0.07$ | $92.41 \pm 0.10$ | $90.72 \pm 0.07$ | $85.60 \pm 0.50$ | $78.15 \pm 0.71$ |
| | MAE | $90.72 \pm 3.37$ | $93.74 \pm 0.06$ | $91.71 \pm 1.88$ | $88.24 \pm 2.11$ | $86.50 \pm 2.16$ |
| | MAE* $\epsilon = 0.5$ | $94.77 \pm 0.03$ | $\mathbf{94.42} \pm 0.02$ | $\mathbf{94.23} \pm 0.05$ | $\mathbf{93.50} \pm 0.11$ | $92.04 \pm 0.12$ |
| Fashion-MNIST | AGCE | $94.39 \pm 0.06$ | $94.24 \pm 0.07$ | $94.01 \pm 0.06$ | $93.35 \pm 0.06$ | $\mathbf{92.28} \pm 0.09$ |
| | genCE | $94.79 \pm 0.06$ | $94.40 \pm 0.06$ | $94.28 \pm 0.02$ | $93.46 \pm 0.07$ | $91.11 \pm 0.15$ |
| ResNet-32 | NCE-AGCE | $92.73 \pm 0.10$ | $92.57 \pm 0.10$ | $92.12 \pm 0.05$ | $91.58 \pm 0.09$ | $90.09 \pm 0.13$ |
| | NF-MAE | $93.92 \pm 0.07$ | $93.74 \pm 0.06$ | $93.71 \pm 0.06$ | $92.84 \pm 0.07$ | $91.75 \pm 0.10$ |
| | NCE-MAE | $93.98 \pm 0.04$ | $93.73 \pm 0.07$ | $93.51 \pm 0.06$ | $93.00 \pm 0.07$ | $91.73 \pm 0.12$ |
| | biTemp | $94.78 \pm 0.03$ | $93.38 \pm 0.03$ | $91.49 \pm 0.09$ | $86.65 \pm 0.50$ | $78.05 \pm 0.47$ |

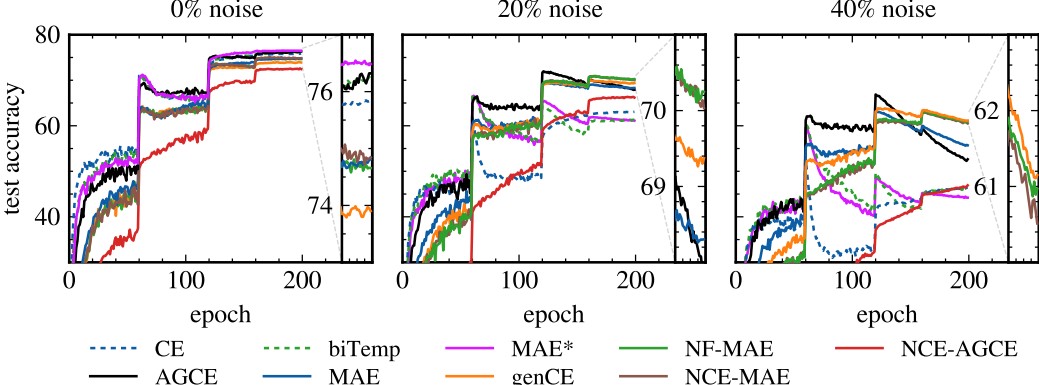

Figure 4: Test accuracy progression on Cifar-100 across training epochs, employing different loss functions under varying label noise levels. Accuracies are averaged over five distinct ResNet network initializations (seeds). Panels are organized by label noise levels: 0% (left), 20% (center), and 40% (right). Insets within each panel offer a closer look at the final 10% of epochs, highlighting subtle differences among the top-performing loss functions.

Table 3: Final test accuracies after training ResNet-34 for 200 epochs on the 100-class dataset Cifar-100 with different loss functions and various amounts of label noise.

| Dataset, Network | Loss | Noise: 0 % | 10 % | 20 % | 40 % | 60 % |
|---|---|---|---|---|---|---|
| | CE | $75.88 \pm 0.15$ | $69.21 \pm 0.12$ | $63.07 \pm 0.15$ | $46.46 \pm 0.24$ | $24.69 \pm 0.68$ |
| | MAE | $6.84 \pm 0.66$ | $5.30 \pm 0.36$ | $4.93 \pm 0.43$ | $2.54 \pm 0.29$ | $1.67 \pm 0.17$ |
| | MAE* $\epsilon = 1.5$ | $74.81 \pm 0.13$ | $72.33 \pm 0.13$ | $68.30 \pm 0.24$ | $55.71 \pm 0.45$ | $35.51 \pm 0.19$ |
| | MAE* $\epsilon = 3.0$ | $\mathbf{76.49} \pm 0.09$ | $69.11 \pm 0.14$ | $61.13 \pm 0.16$ | $44.20 \pm 0.41$ | $26.45 \pm 0.16$ |
| Cifar-100 | AGCE | $76.32 \pm 0.12$ | $\mathbf{73.23} \pm 0.11$ | $67.91 \pm 0.19$ | $52.64 \pm 0.15$ | $32.48 \pm 0.12$ |
| ResNet-34 | genCE | $73.90 \pm 0.17$ | $72.15 \pm 0.09$ | $69.37 \pm 0.20$ | $\mathbf{61.15} \pm 0.10$ | $\mathbf{41.52} \pm 0.64$ |
| | NCE-AGCE | $72.46 \pm 0.09$ | $70.00 \pm 0.30$ | $66.20 \pm 0.35$ | $46.91 \pm 0.69$ | $10.86 \pm 0.42$ |
| | NF-MAE | $74.66 \pm 0.17$ | $72.91 \pm 0.14$ | $\mathbf{70.25} \pm 0.06$ | $\mathbf{60.97} \pm 0.14$ | $25.23 \pm 0.68$ |
| | NCE-MAE | $74.85 \pm 0.10$ | $72.71 \pm 0.07$ | $70.03 \pm 0.10$ | $60.51 \pm 0.10$ | $26.77 \pm 0.81$ |
| | biTemp | $76.19 \pm 0.14$ | $68.40 \pm 0.12$ | $61.36 \pm 0.19$ | $46.15 \pm 0.23$ | $26.67 \pm 0.16$ |

Table 4: Final accuracies on the validation images of the 1000-class WebVision dataset after training Wide-ResNet-101 with different loss functions. If an error is provided, accuracies for five different seeds are averaged. For the losses that do not learn, we report the accuracies of a single training run.

|  | NF-MAE | AGCE-NCE | genCE | CE | MAE* | biTemp | MAE** |
|---|---|---|---|---|---|---|---|
| Top-1 | 1.04 | 7.48 | 13.28 | $62.55 \pm 0.16$ | $58.72 \pm 0.11$ | $\textbf{63.60} \pm 0.14$ | $62.70 \pm 0.07$ |
| Top-5 | 1.17 | 10.08 | 15.39 | $79.63 \pm 0.10$ | $79.68 \pm 0.04$ | $80.66 \pm 0.13$ | $\textbf{81.11} \pm 0.05$ |

relative to CE. It is worth noting that $\epsilon$ is not optimized, but derived from equation Eq. (4), based on the results of the ten-class tasks.

For the case of Cifar-100 with added noise, a different picture emerges: the slower learners display superior generalization. This is attributed to their resistance to overfitting, contrasting starkly with CE, biTemp, and MAE*, which, while exhibiting initial proficiency, witness a rapid decline in test accuracy later on (see Fig. 4). We summarize these results in Tab. 3, where we also demonstrate that the logit bias $\epsilon$ can be used to tune a balance between learning speed and noise-robustness.

In a scenario with 100 classes, the majority of the proposed loss functions still demonstrate good performance, albeit at a marginally reduced learning speed. Their design incorporates "tails" that encompass the logit distribution at initialization. These tails ensure non-zero gradients, enabling the learning of the dataset. However, when scaling to 1000 classes, these tails appear inadequate (see Tab. 4). Here, biTemp loss emerges as the optimal balance, preventing overfitting to noise while still facilitating dataset learning. Though MAE* learns well with the calculated value of the logit bias $\epsilon = 5.2$, removal of the logit bias during testing reveals reduced confidence in the correct label, induced by the elevated $\epsilon$ value during the training phase. Mitigating this can be achieved by adopting smaller values of $\epsilon$ throughout training (denoted as MAE**), as elaborated in the appendix. We observed that the MAE** with a fluctuating logit bias closely parallels the Cross Entropy loss in top-1 accuracy, only slightly lagging behind biTemp. However, in terms of top-5 accuracy, MAE** performs best across all loss functions.

*Limitations*— Our current investigation predominantly focuses on symmetric label noise scenarios. Integrating controlled label noise that favors particular classes would broaden the experimental framework. Even though our theoretical insights indicate that our primary findings should exhibit general applicability across numerous learning tasks, the current scope is confined to image classification challenges. Furthermore, it seems that our present computation of $\epsilon$ based on the number of classes tends to overestimate the required $\epsilon$, which can compromise noise robustness. To address this issue, future theoretical studies could explore a more sophisticated understanding of this phenomenon, while empirical analyses could configure $\epsilon$ as a tunable hyperparameter to circumvent this limitation.

## 5 CONCLUSION

In this study, we analyzed the dynamics of early-stage learning by computing the average backpropagation error, offering quantitative insights into how increasing the class count influences initial learning, especially in the context of bounded loss functions such as MAE. Capitalizing on these insights, we introduced a hyperparameter-independent approach called *logit bias*. This technique realigns the distribution of a newly initialized network, enabling effective learning with the MAE loss even in scenarios with a multitude of classes. Our empirical evidence underscores the effectiveness of this method, with the logit bias enhanced MAE loss demonstrating comparable, if not superior, performance across datasets spanning ten to a thousand classes. Before our approach, such outcomes were largely exclusive to the Cross Entropy or biTemp loss. However, both these methods tend to overfit. In summary, we argue that our method is a first step towards a comprehensive framework - one that allows for noise robust learning, regardless of the number of classes, and without an over-reliance on fine-tuned hyperparameters.

### ACKNOWLEDGMENTS

Acknowledgments hidden for double blind review.

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

# A    APPENDIX

Additional information regarding the manuscript.

## A.1    TRAINING PARAMETERS

Here, we describe the experimental framework used to compare various loss functions.

*Networks*—We train two distinct types of network architectures: (i) residual, convolutional networks utilizing the ResNet architecture (He et al., 2016) with ReLU activation[2], and (ii) multilayer feedforward neural networks, termed MLP1024, comprised of dataset-specific input and output layers, along with three hidden layers of sizes 1024, 512, and 512, also with ReLU activations. The results of the MLP1024 network are only shown in the appendix.

*Preprocessing*—When training ResNet networks, we normalize Cifar-10 images by subtracting the mean $\mu = (0.485, 0.456, 0.406)$ and then dividing by $\sigma = (0.229, 0.224, 0.225)$ for the three color channels. On each training data batch, we further perform a random horizontal flip followed by random cropping to size $32 \times 32$ with a padding of size $4$.

For training ResNet on Fashion-MNIST, we normalize the black and white images using $\mu = 0.286$ and $\sigma = 0.353$. Batches during training are processed with a random horizontal flip followed by random cropping to a size $28 \times 28$ with a padding of size $4$.

The Cifar-100 dataset is normalized using $\mu = (0.507, 0.487, 0.441)$ and $\sigma = (0.267, 0.256, 0.276)$ and the training data is processed with random cropping ($32 \times 32$, padding $4$), followed by a random horizontal flip, and a random rotation by up to $15°$.

Training ResNet networks on WebVision, we normalize the images by subtracting the mean $\mu = (0.485, 0.456, 0.406)$ and then dividing by $\sigma = (0.229, 0.224, 0.225)$ for the three color channels. On each training data batch, we further perform a random horizontal flip followed by random cropping to size $224 \times 224$.

For training MLP1024 networks, we only normalize the images. All test data is normalized identically to the corresponding training data and no further processing is performed.

*Training*—All ResNet networks, are trained using stochastic gradient descent (SGD) with a momentum of $0.9$, a weight decay of $10^{-4}$, and a step-wise learning rate schedule with an initial learning rate of $0.1$. On Fashion-MNIST and Cifar-10 the learning rate is multiplied with a factor of $0.1$ at epochs 100 and 150. For Cifar-100 the learning rate changes at epochs 60, 120, 160 by a factor of $0.2$ and for WebVision we change at epochs 66 and 132 by a factor of $0.1$. The mini-batch size is 128 in all cases but WebVision, where we change it to $400$. For WebVision, we further reduced the training time by only considering the Google images, reducing the number of images from $2.4$ million to a little more than a million.

MLP1024 networks are trained with SGD, with momentum set to $0.95$, a mini-batch size of 32, and no weight decay. We use an exponential learning rate schedule with a decay factor of $0.95$ for each

---

[2]For ResNet networks, we adhere to the `pytorch` implementation described in Refs. Idelbayev (2020); weiaicunzai (2020) for Cifar-10 and 100 respectively, while Wide-ResNet-101 is used directly from the `pytorch` library.

Table 5: Hyperparameters for the different loss functions on various datasets. The parameters are adapted from the respective papers, in the case of WebVision we took the parameters used for Cifar-100.

|          | Fashion-MNIST | Cifar-10 | Cifar-100 | WebVision |
|----------|---------------|----------|-----------|-----------|
| biTemp   | $t_1 = 0.8, t_2 = 1.2$ | $t_1 = 0.8, t_2 = 1.2$ | $t_1 = 0.8, t_2 = 1.2$ | $t_1 = 0.8, t_2 = 1.2$ |
| genCE    | $q = 0.7$ | $q = 0.7$ | $q = 0.7$ | $q = 0.7$ |
| AGCE     | $a = 0.6, q = 0.6$ | $a = 0.6, q = 0.6$ | $a = 1e-5, q = 0.5$ | $a = 1e-5, q = 0.5$ |
| NF-MAE   | $\alpha = 1, \beta = 20$ | $\alpha = 1, \beta = 20$ | $\alpha = 1, \beta = 0.2$ | $\alpha = 1, \beta = 0.2$ |
| NCE-MAE  | $\alpha = 1, \beta = 20$ | $\alpha = 1, \beta = 20$ | $\alpha = 1, \beta = 20$ | $\alpha = 1, \beta = 20$ |
| NCE-AGCE | $\alpha = 1, \beta = 4$ $a = 6, q = 1.5$ | $\alpha = 1, \beta = 4$ $a = 6, q = 1.5$ | $\alpha = 10, \beta = 0.1$ $a = 1.8, q = 3$ | $\alpha = 10, \beta = 0.1$ $a = 1.8, q = 3$ |

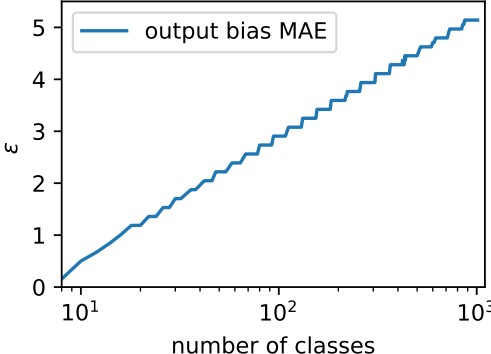

Figure 5: The logit bias value $\epsilon$ for the Mean Absolute Error as a function of the number of classes present in the dataset.

epoch. For the small MLP1024, we additionally perform a grid searches over the initial learning rates.

The loss function specific hyperparameters that were used are shown in Table 5. Here, we adopted the parameters used in the corresponding papers for Cifar-10 and Cifar-100. If not provided otherwise, we used the Cifar-10 parameters for Fashion-MNIST and the Cifar-100 parameters for WebVision.

Networks are initialized with zero biases and random Xavier-uniform entries (Xavier Glorot & Yoshua Bengio, 2010) for the MLP networks, while the ResNet architectures use Kaiming Normal weights (He et al., 2015). We train the networks using five different initialization seeds. For robust results when comparing network performance, we report the mean of the accuracies over the five different network realizations, along with the corresponding errors of the mean.

### A.2 CHOOSING THE RIGHT LOGIT BIAS

In Eq. (4) we give a criterion that defines how $\epsilon$ may be chosen given the number of classes in the dataset, and an additional constant $C$ which is set to $0.15$ in the whole manuscript. In Figure 5 we show the dependency of $\epsilon$ on the number of classes using this equation.

On the WebVision dataset we found that having a rather large $\epsilon$ leads to low confidence predictions on the test set, as the network is used to the logit bias. To counteract this issue, we proposed to wean the network by also including epochs with a lower $\epsilon$ value. This was done by taking 6 discrete $\epsilon$ values of $[5.2, 4.8, 4.3, 3.7, 3.0, 2.3]$ to iterate through during training. Training starts with $\epsilon = 5.2$ and after 11 epochs, the next smaller $\epsilon$ is taken. After 66 epochs, the smallest $\epsilon$ is reached and the next epoch starts again with $\epsilon = 5.2$. The reason for a decay that repeats after 66 epochs is the change in the learning rate after 66 and 132 epochs respectively. Hence, each learning rate gets an identical schedule. As this cycle completes after 198 epochs we decided to only train for this amount of epochs in these cases.

In general we believe that a dynamically tuned $\epsilon$ could further improve performance even for datasets with fewer classes than WebVision. If the network gains confidence in a specific example in the later stages of training, the network bias either prevents full memorization of the example as in the case of 1000 classes or even allows an example with the wrong label to be learned even though the network was already fairly confident against it. We have not included this in the present manuscript, since this approach could be considered to be fine-tuning of $\epsilon$, such that to ensure a fair comparison with other loss functions we would have to fine-tune the respective parameters as well, substantially increasing the compute time for numerical experiments.

The problem of having to actively find the correct $\epsilon$ schedule could principally be mitigated by designing a different loss function that uses our insight into the early stages of learning. By choosing the derivative of the loss function to extend to the middle of the initial distribution independent of the class number we could most likely omit any schedule. However, this is a topic for future research.

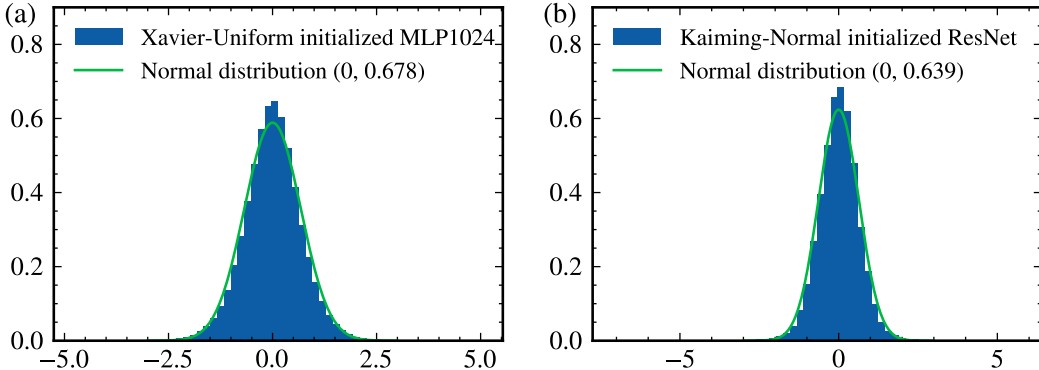

Figure 6: Logit distribution $p_z$ for (a) the fully connected network MLP1024 and (b) the ResNet-34 architecture. The histogram displays the empirical logits for different network initializations given the first images of the Cifar dataset. The green curve displays a normal distribution with a mean of zero. The variance of the normal distribution is set to the variance of the logit distribution.

### A.3 Empirical output of a freshly initialized neural network

In the main text, we derive that the logit distribution of a freshly initialized neural network $p_z$ follows a normal distribution with a mean of zero if the weights are initialized with a mean of zero. The standard deviation depends on the distribution which was used to initialize the weights. This is verified in Figure 6 where panel (a) shows the empirical logits of 500 Xavier-uniform initialized MLP1024 networks when given the first 150 Cifar-10 images. Panel (b) shows the logits of 20 different ResNet-34 networks initialized with Kaiming-Normal distributed numbers given the first 300 images of the Cifar-100 dataset.

### A.4 Estimating hyperparameters of other loss functions

Other loss functions often use hyperparameters to adjust for different learning scenarios. The various parameters used in the manuscript are summarized in A.1 and show that their behaviour can be highly non trivial. To reduce hyperparameters we propose to instead use the overlap shown in Fig. 2 to adjust these parameters. Using parameters which are known to work with a specific number of classes, one computes the value of the backpropagation error $\langle \delta_k \rangle (z_k)$ at the position $z_k = 0$ for this number of classes. When changing the number of classes, the hyperparameters are adjusted until $\langle \delta_k \rangle (0)$ coincides with the value of the baseline for which it is known that learning works well.

As an example, we consider genCE where we use that for $C = 100$ classes the value $q_{100} = 0.7$ has been established and compute the value of the backpropagation error $\langle \delta_k \rangle (0)$ to be around $-0.279$. For $C = 1000$ classes, we find that $q_{1000} = 0.48$ exactly mimics this behavior.

### A.5 Using the logit bias to boost generalized cross entropy

We speculate that the logit bias is capable of boosting the performance of other bounded losses than MAE. When training genCE with the calculated value $\epsilon = 2.4$ on the WebVision dataset, we find that the performance increases from $13.3\%$ to $62.4\%$.

### A.6 Empirical results on a different architecture

To train MLP1024 networks on the Cifar-10 and Fashion-MNIST datasets, we perform a grid search over nine values of initial learning rates in the absence of label noise. The value with the best average accuracy out of five runs for each learning rate is then used for training in the presence of varying degrees of label noise. The results for the learning rate optimization are shown in Table 6, where values in bold font mark the learning rate used for the simulations with noise.

Table 6: Learning rate optimization for various loss functions on the clean dataset for MLP1024 networks. The test accuracies in boldface indicate the learning rate for which results are presented in the main manuscript. MAE* has $\epsilon = 0.5$

| learning rate | 0.0005 | 0.0008 | 0.001 | 0.003 | 0.005 | 0.008 | 0.01 | 0.03 | 0.05 |
|---|---|---|---|---|---|---|---|---|---|
| **Cifar-10** | | | | | | | | | |
| CE | 54.42 ± 0.06 | 55.01 ± 0.10 | **55.70** ± 0.13 | 55.41 ± 0.15 | 55.03 ± 0.09 | 54.65 ± 0.10 | 54.44 ± 0.13 | 10.00 ± 0.00 | 10.00 ± 0.00 |
| MAE | 49.44 ± 1.11 | **49.86** ± 1.13 | 49.18 ± 1.82 | 43.03 ± 1.26 | 22.80 ± 2.81 | 11.37 ± 0.97 | 10.92 ± 0.92 | 10.00 ± 0.00 | 10.00 ± 0.00 |
| MAE* $\epsilon = 0.5$ | 52.98 ± 0.13 | 53.84 ± 0.05 | 54.20 ± 0.09 | **55.09** ± 0.14 | 42.67 ± 3.46 | 10.35 ± 0.35 | 12.20 ± 0.97 | 10.00 ± 0.00 | 10.00 ± 0.00 |
| genCE | 52.54 ± 0.16 | 53.54 ± 0.16 | 53.91 ± 0.07 | 54.99 ± 0.18 | **55.21** ± 0.13 | 55.02 ± 0.26 | 54.57 ± 0.07 | 10.00 ± 0.00 | 10.00 ± 0.00 |
| NF-MAE | 53.67 ± 0.15 | 53.88 ± 0.10 | **53.98** ± 0.13 | 16.48 ± 2.93 | 14.92 ± 1.88 | 10.00 ± 0.00 | 10.83 ± 0.83 | 10.46 ± 0.46 | 10.00 ± 0.00 |
| NCE-MAE | 53.82 ± 0.04 | **53.77** ± 0.10 | 53.49 ± 0.21 | 16.61 ± 0.81 | 14.12 ± 1.10 | 13.09 ± 1.04 | 11.43 ± 1.43 | 10.00 ± 0.00 | 10.00 ± 0.00 |
| biTemp | 53.17 ± 0.15 | 54.23 ± 0.07 | 54.69 ± 0.18 | 55.89 ± 0.22 | **56.02** ± 0.24 | 55.21 ± 0.17 | 54.97 ± 0.18 | 50.76 ± 0.25 | 28.66 ± 8.11 |
| **Fashion-MNIST** | | | | | | | | | |
| CE | 89.39 ± 0.06 | 89.63 ± 0.08 | 89.64 ± 0.08 | 90.11 ± 0.05 | **90.20** ± 0.03 | 89.97 ± 0.05 | 90.08 ± 0.06 | 89.19 ± 0.10 | 10.00 ± 0.00 |
| MAE | 81.24 ± 2.24 | **84.83** ± 2.32 | 82.53 ± 2.60 | 82.11 ± 2.21 | 84.34 ± 2.00 | 79.61 ± 2.65 | 72.00 ± 1.88 | 10.03 ± 0.03 | 12.00 ± 1.79 |
| MAE* $\epsilon = 0.5$ | 88.72 ± 0.05 | 88.52 ± 0.95 | 88.52 ± 0.95 | **89.55** ± 0.03 | 88.52 ± 0.95 | 89.43 ± 0.06 | 87.55 ± 0.52 | 10.00 ± 0.00 | 10.01 ± 0.01 |
| genCE | 88.54 ± 0.05 | 88.82 ± 0.03 | 88.93 ± 0.04 | 89.57 ± 0.05 | 89.60 ± 0.08 | 89.75 ± 0.06 | **89.76** ± 0.06 | 17.00 ± 1.60 | 11.06 ± 0.95 |
| NF-MAE | 88.73 ± 0.05 | 89.06 ± 0.04 | **89.13** ± 0.03 | 88.30 ± 0.03 | 30.28 ± 4.61 | 20.34 ± 7.53 | 17.47 ± 1.70 | 10.00 ± 0.00 | 12.36 ± 1.57 |
| NCE-MAE | 88.82 ± 0.03 | 89.05 ± 0.03 | **89.14** ± 0.03 | 88.31 ± 0.05 | 43.06 ± 8.93 | 10.00 ± 0.00 | 13.94 ± 2.16 | 15.95 ± 2.17 | 11.87 ± 1.66 |
| biTemp | 89.08 ± 0.03 | 89.33 ± 0.10 | 89.50 ± 0.06 | 89.97 ± 0.06 | 90.13 ± 0.06 | 90.16 ± 0.07 | **90.19** ± 0.07 | 89.93 ± 0.06 | 89.44 ± 0.09 |

Table 7: Final test accuracies for training MLP1024 networks on the ten-class datasets Cifar-10 and Fashion-MNIST with different loss functions and various amounts of label noise. For the optimized learning rates see Tab. 6.

| Dataset, Network | Loss | Noise: 0 % | 10 % | 20 % | 40 % | 60 % |
|---|---|---|---|---|---|---|
| | CE | **55.70** ± 0.13 | 50.56 ± 0.10 | 46.20 ± 0.31 | 36.90 ± 0.17 | 27.51 ± 0.35 |
| | MAE | 49.86 ± 1.13 | 49.08 ± 0.91 | 48.20 ± 1.04 | 45.31 ± 0.93 | 40.27 ± 1.06 |
| Cifar-10 | MAE* $\epsilon = 0.5$ | 55.09 ± 0.14 | **53.83** ± 0.15 | **53.18** ± 0.20 | 49.30 ± 0.08 | 42.29 ± 0.13 |
| | genCE | 55.21 ± 0.13 | **54.07** ± 0.22 | 52.72 ± 0.14 | 47.94 ± 0.13 | 36.70 ± 0.26 |
| MLP1024 | symCE | 54.13 ± 0.14 | 49.03 ± 0.17 | 44.30 ± 0.13 | 34.50 ± 0.19 | 25.23 ± 0.15 |
| | NF-MAE | 53.98 ± 0.13 | 53.20 ± 0.15 | 52.00 ± 0.08 | **49.90** ± 0.18 | **45.43** ± 0.17 |
| | NCE-MAE | 53.77 ± 0.10 | 53.10 ± 0.05 | 52.18 ± 0.12 | **49.62** ± 0.24 | 44.88 ± 0.21 |
| | biTemp | **56.02** ± 0.24 | 51.05 ± 0.11 | 46.58 ± 0.18 | 37.35 ± 0.21 | 27.28 ± 0.22 |
| | CE | **90.20** ± 0.03 | 86.05 ± 0.04 | 79.54 ± 0.13 | 63.04 ± 0.34 | 45.67 ± 0.39 |
| | MAE | 84.83 ± 2.59 | 81.25 ± 1.44 | 82.00 ± 2.02 | 83.70 ± 1.58 | 80.17 ± 2.60 |
| Fashion-MNIST | MAE* $\epsilon = 0.5$ | 89.55 ± 0.03 | 89.48 ± 0.05 | **89.09** ± 0.07 | **88.03** ± 0.07 | 85.63 ± 0.12 |
| | genCE | 89.76 ± 0.07 | **89.49** ± 0.05 | **89.09** ± 0.10 | 87.70 ± 0.12 | 80.86 ± 0.19 |
| MLP1024 | symCE | 89.98 ± 0.07 | 88.06 ± 0.04 | 84.05 ± 0.13 | 69.95 ± 0.37 | 51.34 ± 0.20 |
| | NF-MAE | 89.13 ± 0.03 | 88.99 ± 0.04 | 88.62 ± 0.06 | **87.86** ± 0.10 | **86.26** ± 0.08 |
| | NCE-MAE | 89.14 ± 0.03 | 88.95 ± 0.06 | 88.59 ± 0.05 | 87.79 ± 0.08 | **86.24** ± 0.06 |
| | biTemp | **90.19** ± 0.08 | 86.46 ± 0.11 | 80.33 ± 0.15 | 64.11 ± 0.37 | 45.77 ± 0.29 |

The final results when training MLP1024 Networks on Cifar-10 and Fashion-MNIST are summarized in Tab. 7, where we find that MAE* performed best under all loss functions.

## A.7 BACKPROPAGATION ERROR OF BOUNDED LOSSES

For completeness, we provide the backpropagation errors $\delta_n$ for the active-passive losses and AGCE-NCE in this section. As in the main text, $k$ denotes the index of the non-zero entry in the corresponding one-hot encoded label, i.e. $k = \mathrm{argmax}(\boldsymbol{y})$.

The active passive losses are defined as

$$\mathrm{NF} - \mathrm{MAE} = \alpha \, \frac{\log\left((1-a_k)^{0.5} a_k\right)}{\sum_i \log\left((1-a_i)^{0.5} a_i\right)} + \beta \, \mathrm{MAE}(\boldsymbol{a}, \boldsymbol{y}) \,, \tag{5}$$

$$\mathrm{NCE} - \mathrm{MAE} = \alpha \, \frac{\log(a_k)}{\sum_i \log(a_i)} + \beta \, \mathrm{MAE}(\boldsymbol{a}, \boldsymbol{y}) \,. \tag{6}$$

We are interested in the error $\delta_n = \partial_{z_n} \mathcal{L}$ where $a_i = \exp(z_n)/\sum_i \exp z_i$. The derivative of the mean absolute error (MAE) is given in the main text as $2a_k(a_k - \delta[n-k])$. We further find that

$$\frac{\partial}{\partial z_n} \frac{\log\left((1-a_k)^{0.5}a_k\right)}{\sum_i \log\left((1-a_i)^{0.5}a_i\right)} =$$

$$\frac{(\delta[n-k]-a_n)\left(1 - \frac{a_k}{2-2a_k}\right)\left(\sum_i \log((1-a_k)^{0.5}a_k)\right) - \left(\sum_i(\delta[n-i]-a_n)(1-\frac{a_i}{2-2a_i})\right)\log((1-a_k)^{0.5}a_k)}{\left(\sum_i \log((1-a_i)^{0.5}a_i)\right)^2},$$

$$\frac{\partial}{\partial z_n} \frac{\log(a_k)}{\sum_i \log(a_i)} = \frac{(\delta[n-k]-a_n)\sum_i \log(a_i) - \log(a_k)(-ca_n+1)}{\left(\sum_i \log(a_i)\right)^2}.$$

(7)

For the AGCE-NCE we simply combine the already found derivatives:

$$\frac{\partial}{\partial z_n} \frac{\log(a_k)}{\sum_i \log(a_i)} + [(b+1)^q - (b+a_k)^q]/q =$$

$$\frac{(\delta[n-k]-a_n)\sum_i \log(a_i) - \log(a_k)(-ca_n+1)}{\left(\sum_i \log(a_i)\right)^2} + a_k(a_n - \delta[n-k])(b+a_n)^{q-1}$$

(8)

A visualization of these formulas is provided in the main manuscript.

