# OpenReview forum: "Bounded Loss Robustness: Enhancing the MAE Loss for Large-Scale Noisy Data Learning"
_ICLR.cc/2024/Conference — Submitted to ICLR 2024_

### Official Review · Reviewer_kScR · 2023-10-20

**Soundness:** 2 fair
**Presentation:** 3 good
**Contribution:** 3 good
**Rating:** 5
**Confidence:** 3

**Summary:**

This paper explored the learning behavior of MAE (one of symmetric loss functions), i.e. the limited overlap between the network output at the initial phase and non-zero derivative regions of the loss function. For tackling this issue, the paper introudced 'logit bias' to restore the overlap, and enabled MAE to learn on datasets with noisy labels. Extensive experiments on various datasets show the effectiveness of the proposed method.

**Strengths:**

1. The paper provided detailed  analysis of proposed 'logit bias' from both theoretical (Eq.1 -- Eq.4) and experimental point of view (comparing with various loss functions).
2. The author's writing is very good, and the entire paper is relatively easy to understand.
3. Simple algorithm, easy to follow as only one additional parameter (ϵ) are needed.
4. The experimental results are reliable and sufficient to verify the effectiveness of this method.
5. The last paragraph of Section 4 discusses some limitations.

**Weaknesses:**

1. The proposed method dosen't achieve SOTA accuracy in Table 2, 3, 4. The method didn't show dominant advantages over other loss functions, further explorations are expected. (current improvements are not very significant)
2. The estimation of "logit bias" is still empirical for tasks or number of classes, which is not easy to configure it for different tasks.

**Questions:**

1. "thus laying the foundation for a universal classification framework." Can this work support this strong conclusion?
2. one paper relates to this work, would you please give some comments (comparison): IMAGE for Noise-Robust Learning: Mean Absolute Error Does Not Treat Examples Equally and Gradient Magnitude's Variance Matters,Published at ICLR 2023 Workshop on Trustworthy and Reliable Large-Scale ML Models.

---

> ### Author Response · Authors · 2023-11-14
>
> We thank the Reviewer for the feedback and the interesting reference.
>
> The Reviewer's questions and weaknesses scrutinize our concept for choosing $\epsilon$. As a first step, our manuscript analyzes the overlap of the derivative of bounded loss functions with the logit values in the early learning phase. In this phase, the weights are still random such that it is possible to explicitly calculate how this overlap behaves as a function of the number of classes in the dataset. As more classes lead to a vanishing overlap of the two regions, we propose to revert this by shifting the logit encoding the correct class.
> However, we do not use a heuristic value for the logit bias that must be tuned for each dataset,  but precisely calculate the $\epsilon$ value necessary to ensure optimal learning for differing class numbers using Eq. (4) of the manuscript.
>
> We argue that previous work used hyperparameters that must be tuned as soon as the number of classes in the dataset is changed. This greatly differs from our approach which provides a condition for choosing $\epsilon$. For example, if you were to train a network with 150 classes, we would suggest $\epsilon=3.25$ without training a neural network once. Fig. 5 in the appendix illustrates this behavior.
> We believe that future work could benefit from our analysis by calculating how hyperparameters should be adjusted as a function of the number of classes in the dataset, therefore providing a more general framework that must not be fine-tuned for each individual dataset. In this spirit, we formulated our conclusion which the Reviewer finds too strong.
> However, we accept the Reviewer's criticism and have somewhat softened our conclusion to "thus being a first step towards a class count independent framework." in the updated version of the manuscript.
>
>
> Given that we calculate the parameter $\epsilon$ and do not fine-tune it, we believe that even though our method is not superior in all cases, the fact that it is among the best on all data sets is a considerable achievement. This is true even more since it was believed that the MAE loss is incapable of training on data sets with many classes.
>
> We thank the Reviewer for pointing out the interesting reference "IMAGE for Noise-Robust Learning: Mean Absolute Error Does Not Treat Examples Equally and Gradient Magnitude's Variance Matters", published in ICLR 2023.  The reference takes a similar first approach by analyzing the gradients with respect to the logits. By re-weighting the gradients using an exponential function, the MAE can train effectively - however at the price of changing the  MAE loss fundamentally. As a consequence, this approach has several drawbacks: the modified MAE loss loses the theoretically derived guarantees for noise robustness, as the modified loss is neither bounded nor symmetric. In addition, this approach introduces a hyperparameter $T$ which needs to be tuned.
>
> Besides these theoretical and practical considerations, we find that the final test accuracies in the presence of label noise are significantly better for MAE with logit bias as compared to the re-weighted gradient approach. We use the ResNet-32 architecture for Cifar-10 and ResNet-34 for Cifar-100 while the reference uses ResNet-20 and ResNet-56 on Cifar-10. When taking the better of their two final test accuracies, they achieve $84.0\\%$ on Cifar-10 at $40\\%$ label noise compared to our $87.3\\%$. For Cifar-100 they used ResNet-44 and found $63.4\\%$ compared to our $68.3\\%$ test accuracy for $20\\%$ noise. For $40\\%$ noise, we found $55.7\\%$ while their final accuracy is $54.7\\%$. Only in one case, $60\\%$ noise on Cifar-100, their results were better with an accuracy of $43.9\\%$ compared to our $35.5\\%$.
>
> Considering that the above reference claims SOTA results while our method seems to perform better while also being more straightforward to implement, we believe that our paper is a worthy contribution to ICLR and kindly ask the Reviewer to reconsider the assessment of our work.

---

> > ### Comment · Reviewer_kScR · 2023-11-21
> >
> > Thank you very much for your responses to my questions. These discussions can address my concerns.
> >
> > In summary, I agree that the paper provided theoretical analysis and conducted experiments to validate the proposed MAE*.
> >
> > However, for top-level conferences, the paper still has some weaknesses.
> > - This work looks like intuitively observing experimental results and then providing theoretical explanations. This is a reasonable research method, however, this explanation requires sufficient experiment validation. (I do not think the current experiment results are convincing yet)
> > - This paper can also be accepted if the proposed loss function can achieve SOTA in at least one task across various datasets, however, the current experimental results have not shown superiority. (Reviewer cujv has also pointed out that empirical improvements are trivial. And, Reviewer jRF4 pointed out benchmarks are not sufficient.)
> > - From a theoretical perspective, redesigning the loss function by introducing constants is not novel enough for me. I listed some references in face recognition domain:
> > 1) Deng, Jiankang, J. Guo and Stefanos Zafeiriou.“ArcFace: Additive Angular Margin Loss for Deep Face Recognition.” 2019 IEEE/CVF Conference on Computer Vision and Pattern Recognition (CVPR) (2018): 4685-4694.
> > 2) Wang, H., Yitong Wang, Zheng Zhou, Xing Ji, Zhifeng Li, Dihong Gong, Jin Zhou and Wei Liu. “CosFace: Large Margin Cosine Loss for Deep Face Recognition.” 2018 IEEE/CVF Conference on Computer Vision and Pattern Recognition (2018): 5265-5274.
> >
> > Based on the above weaknesses, I think that the current manuscript does not meet the standards of ICLR. Therefore, I keep my rating.

---

### Official Review · Reviewer_jRF4 · 2023-10-27

**Soundness:** 3 good
**Presentation:** 3 good
**Contribution:** 2 fair
**Rating:** 3
**Confidence:** 4

**Summary:**

This paper aims to handle label noise. They observed that although bounded losses exhibit robustness against label noise, they suffer from serious underfitting. Taking MAE example, the authors explore its learning behavior. Motivated by this, they propose a new method called logit bias to address the underfitting.

**Strengths:**

1. This paper is well-organized and easy to follow.
2. This paper is well-motivated. Specifically, the authors analyze the learning behavior of MAE at the early stage of training process, revealing the reason why MAE suffers from underfitting.
3. Based on the above analysis, the author proposed logit bias, which is easy to implement and really helps to alleviate underfitting in some cases based on their experimental results.

**Weaknesses:**

In my humble opinion, the contributions of this paper seem limited and do not achieve the bar of ICLR.
1. It is widely known that bounded loss such as MAE suffers from underfitting, an early work [1] also performed gradient analysis to explain this phenomenon. Although the analysis in this paper is somewhat different from the early work, I don't think it provides enough new insights.
2. The proposed method "logit bias" seems too simple. It is definitely okay if it is effective enough, unfortunately, it is not.
3. The current experimental results are not sufficient to demonstrate the effectiveness of the proposed method. First, most experiments are performed on symmetric label noise, I wonder if the proposed method can also handle asymmetric noise. I know that Webvision contains asymmetric noise, but its noise rate is relatively low. Moreover, even for symmetric label noise, the proposed method lags behind some previous methods such as genCE in many cases. Finally, I noticed that the authors claim that genCE has no hyper-parameter. I guess that they set $q$ of GCE to 0.7 by default. However, in my experience, the performance of genCE can improve remarkably if we elaborately adjust $q$. For instance, if we set $q$ to 0.5 or a smaller value, genCE might outperform other robust losses on WebVision. In fact, considering that MAE* has one hyper-parameter, it is unfair to freeze $q$ of genCE.

Reference

[1] Generalized cross entropy loss for training deep neural networks with noisy labels, NeurIPS 2018.

**Questions:**

Please see above.

---

> ### Author Response · Authors · 2023-11-14
>
> We thank the reviewer for their feedback.
>
>
> Weakness 1:
>
> We agree that it is common knowledge that the MAE loss suffers from underfitting. However, we politely disagree with the statement that our work does not provide deeper insights than Ref. [1]. While Ref. [1] qualitatively discusses the problem of small gradients for outliers with the conclusion that the loss function must hence be altered, our work quantifies how this effect depends on the number of classes in the early stages of learning.
>
> This allows us to change perspective and adjust the gradients accordingly,  instead of changing the loss function. Furthermore, Ref. [1] adjusts a hyperparameter with the general idea in mind that the parameter should be chosen "sufficiently small" to allow for learning. It is hence a heuristic loss function that must be tuned using experience. In contrast, our method has a parameter that can be calculated as a function of the number of classes.
> Beyond introducing the concept of logit bias, we would like to argue that our framework (i.e. considering the overlap between the distribution of logits and the range of non-zero error) can be helpful in determining the value of the hyperparameter $q$ of Ref. [1] based on the number of classes in the dataset, without running a grid search.
>
>
> Weakness 2:
>
> We politely disagree with the Reviewer's judgment that our method is "too simple". In our view, a simple method is preferable to a complex method as long as it performs as well as the complex method. With regards to the Reviewer's judgment that the "logit bias" is not effective enough, we showed that all of the proposed noise-robust loss functions struggled on one of the three data sets, while the MAE loss with a straightforward modification of the bias value remained functional. This was possible without tuning hyperparameter as the value of $\epsilon$ was determined by using Eq. (4). In addition,  we would like to argue that many of the more complex approaches to bounded loss functions, involving several hyperparameters, are finetuned to a specific problem while a "simple" method is able to perform well in many situations.
>
>
> Weakness 3:
>
> As the Reviewer states, the dataset WebVision contains asymmetric noise and we do not find that this hurts the performance of our loss function. This is further supported by the theoretical results of Ref. [2], where it is established that the MAE is tolerant to asymmetric noise under general conditions.
>
> Furthermore, we do not claim that genCE has no hyperparamers but explicitly write that we use genCE with a value of $q=0.7$ as suggested in the original paper [1]. We agree that tuning the parameter $q$ could improve the performance,  but as argued above,  there is a difference between setting the hyperparameter $q$ vis-a-vis using the $\epsilon$ value computed from Eq. (4). However, we are happy to include results for the suggested value $q=0.5$ in the final version of the manuscript.
>
> In light of the above arguments, we kindly ask the Reviewer to reconsider the assessment of our work.
>
> Reference:
> [2] Robust loss functions under label noise for deep neural networks, AAAI 2017

---

> > ### Comment · Reviewer_jRF4 · 2023-11-19
> > **Response to authors**
> >
> > Thanks for your time and labor in addressing my concern. However, I still decided to maintain my score.
> >
> > I completely agree that this paper provides something different from the analysis in [1]. However, their conclusion that "underfitting depends on the number of classes" is widely known in the community. Besides, I think the theoretical analysis is not deep enough, it is more like an intuitive explanation. I just list some previous works [2,3,4] which I think have brought good theoretical contributions.
> >
> > Although the theoretical contribution is limited, it is definitely okay if the empirical contribution is good. Unfortunately, the current manuscript cannot achieve the bar of ICLR. Firstly, the high-level idea that improving robustness against label noise by adjusting gradients is not novel. Secondly, the baselines are somewhat old and benchmarks are not sufficient (All label noises are symmetric except for WebVision). Thirdly, as also pointed out by Reviewer cujv, the empirical improvement is trivial.
> >
> > In my humble opinion, "simple but effective" is preferable, only "simple" may be equivalent to "trivial".
> >
> > In Table 1, the author claimed that genCE has 0 parameter, which might mean that genCE has no hyper-parameter.
> >
> > [1] Generalized cross entropy loss for training deep neural networks with noisy labels, NeurIPS 2018.
> >
> > [2] When optimizing f-divergence is robust with label noise. ICLR 2021.
> >
> > [3] To smooth or not? when label smoothing meets noisy labels. ICML 2022.
> >
> > [4] Learning from Noisy Pairwise Similarity and Unlabeled Data. JMLR 2022.

---

> > > ### Author Response · Authors · 2023-11-20
> > >
> > > We thank the Reviewer for their response.
> > >
> > > We would like to comment on the critique of the Referee that our theoretical analysis was not deep enough and show below why we think that our method is a "good theoretical contribution".
> > >
> > > We argued before that our framework can be used to determine hyperparameters in loss functions other than MAE, which were previously chosen on a dataset-to-dataset basis using empirical tests. For this, we perform a calculation based on Eq. (3) to determine the parameter $q$ in genCE as a function of the number of classes. We use that for $C=100$ classes the value $q_{100}=0.7$ has been established and compute the value of the backpropagation error $\langle\delta_k\rangle (z_k)$ at the position $z_k=0$. For $C=1000$ classes, we then solve for $q$ such that this value of the backpropagation error remains unchanged, for which we find $q_{1000}=0.48$. This is close to the value of $q=0.5$ that was suggested by the Referee for the WebVision dataset, which we assume was based on empirical results or the intuition of the Referee. To emphasize this point, we also added a discussion of this topic to the Appendix.
> > >
> > > As a second test for the generality of our approach, we also added a logit bias to genCE with $q=0.7$ and trained on the WebVision dataset. We find that adding the logit bias $\epsilon=2.4$ that is obtained from Eq. (4), is already sufficient to enable the genCE loss to achieve $62.4\%$ on a single run, which is a huge improvement over the $13.3\%$ that are obtained for $q=0.7$ without a logit bias.
> > >
> > > We hope that we were able to demonstrate that our method is more general than just contributing a new benchmark loss function. We would also like to emphasize that our method significantly outperforms another one proposed at ICLR 2023 (mentioned by Reviewer kScR), which claimed SOTA results, such that we believe that our benchmarks are far from outdated but rather compare to recent improvements in the field.

---

### Official Review · Reviewer_cujv · 2023-11-01

**Soundness:** 3 good
**Presentation:** 2 fair
**Contribution:** 4 excellent
**Rating:** 5
**Confidence:** 4

**Summary:**

In this study, the authors analyzed the dynamics of early-stage learning by computing the average backpropagation error, providing quantitative insights into how increasing the class count influences initial learning, especially in the context of bounded loss functions such as Mean Absolute Error (MAE). They introduced a hyperparameter-independent approach called "logit bias," which realigns the distribution of a newly initialized network, enabling effective learning with MAE loss even in scenarios with a multitude of classes. Empirical evidence demonstrates the effectiveness of this method, with logit bias enhanced MAE loss showing comparable or superior performance across datasets spanning ten to a thousand classes. This is significant as such outcomes were previously largely exclusive to Cross Entropy or biTemp loss, which tend to overfit. The authors argue that their method is a first step towards a comprehensive framework that allows for noise-robust learning, regardless of the number of classes, and without an over-reliance on fine-tuned hyperparameters.

**Strengths:**

1. This paper offers a novel insight into the underfitting phenomenon observed in some robust loss functions, pinpointing the discord between the non-zero range of the average error and the logit distribution of a newly initialized network as the primary culprit.

2. The approach of this paper is simplicity and efficiency, requiring only a single parameter, ϵ, which is directly determined by the number of classes.

3. The paper provides clear details about the experimental setup, allowing for reproducibility and further exploration by other researchers.

**Weaknesses:**

1. The notations are not clearly defined; for example, the meaning of $\delta_{nj}$ is unclear.

2. This method is only designed for MAE, could it be expanded to help other robust losses?

3. The empirical improvement is trivial. In the dataset CIFAR100 and with Resnet-34, it only achieves the state-of-the-art for clean data.

**Questions:**

1. In figure 1, the label for x-axis is "pre activation $z_k$", but there is no information about the activation function, I wonder what is the specific meaning of $z_k$, $a_j$ and $\delta_{nj}$.

2. How to get the output error of $\delta_n$ in Tabel 1? Is that related to activation function?

------
I acknowledge that I have read the response of the authors. However, I am not convinced by the contribution of this work. Therefore, I tend to keep my score as 5.

---

> ### Author Response · Authors · 2023-11-14
>
> We thank the Referee for the valuable feedback.
>
> Question 1:
>
> We thank the Referee for the comment that our notation which uses $\delta_n$ to denote the error in the backpropagation equations and $\delta_{ij}$ to denote the Kronecker delta can be confusing. In the updated Manuscript, we have modified the notation and now denote the Kronecker delta as $\delta[i,j]$ and explicitly state its definition.
>
> The symbol $z_k$ denotes the logit at the neuron position $k$ while $a_j$ denotes the neuron activation at position $j$ after the activation function is applied, i.e, $a_k = \exp(z_k) / \sum_i   \exp(z_i)$ for the softmax activation function.
> In the updated version we explicitly state the definitions for these quantities in the paragraph below Eq. (2), where they occur for the first time.
>
>
> Following this notation, the "pre-activation $z_k$" in Fig. 1 is the logit at position $k$. We use Eq. (3) to calculate the error for each of the logit values, where we assume $a_k = \exp(z_k) / \sum_i  \exp(z_i)$, i.e., the softmax activation, which is more prominently mentioned in the updated version.
>
> Question 2:
>
> Regarding the errors in Tab. 1, we use Eq. (2) to calculate these values. The assumed activation in the final layer is the softmax activation function. As mentioned, we tried to make this clearer in an updated version.
>
> Remarks:
>
> As for the improvements on the Cifar-100 dataset, we agree that this is not a new state-of-the-art benchmark outperforming Cross-Entropy in the case of clean data. However, MAE with logit bias keeps a large degree of robustness against cross-entropy and clearly outperforms it when label noise is introduced. In addition, we would like to argue that achieving state-of-the-art results with the MAE loss on clean data is still impressive, considering that none of the training hyperparameters are optimized for the MAE loss.
>
> As for improving the performance of other loss functions besides MAE, we believe that our analysis could be helpful both directly by applying an output bias to facilitate learning, but also indirectly: By providing a quantitative metric (the overlap between the logit distribution and the error) for how learning is impaired when the number of classes increases, the hyperparameters of other loss functions could potentially be adjusted based on the number of classes. This would potentially greatly improve their reliability and make them easier to use in practice.
>
> In view of our above answers, we believe that our manuscript is a worthy contribution to ICLR and kindly ask the Reviewer to reconsider their assessment of our work.

---

### Official Review · Reviewer_n8a1 · 2023-11-07

**Soundness:** 3 good
**Presentation:** 3 good
**Contribution:** 3 good
**Rating:** 8
**Confidence:** 3

**Summary:**

This paper explores the learning behavior of bounded loss functions and proposes a modified version of MAE (adding logit-bias), addressing its under-fitting issue, especially when used in scenarios with a large number of classes. Both theoretical and numerical analyses demonstrate the picture behind the proposed logit-bias. Experiments show the proposed method's effectiveness and superiority over other noise robust losses.

**Strengths:**

- the paper is well-written, and the idea is clearly presented.
- the observation of the overlapping between the distributions of the averaged error and the logit and its implication on network learning is keen and illuminating.
- the proposed logit-bias is simple yet effective.

**Weaknesses:**

- the study of learning behavior still needs to be completed. It only considers the learning at the initialization stage.
    - for example, in Fig. 4, the test acc using MAE* degrades in the late stage of training when the noise is present. Why does this happen? It would be interesting to investigate the learning behavior during the full training phase.
- it would be desirable to consider more types of noise, e.g., skewed label noise, feature-dependent noise, etc., and more network architectures.
- the way of choosing optimal $ \epsilon $ needs to be explored more.

**Questions:**

- should we not consider the two temperatures as parameters for the Bi-Tempered loss?
- how would the overlapping state change during training? would a dynamically tuned $\epsilon$ helpful? would you consider other strategies for choosing $\epsilon$?
- is it possible, based on your observation, to design an initialization method that can also improve the overlapping?
- as had been noticed in the paper, training with the logit-bias may introduce a kind of inductive bias to the resulting network, and this issue is addressed in a heuristic way by using small $\epsilon$ values. However, a smaller $\epsilon$ would have reduced its ability to restore overlapping. Could you elaborate on how this inductive bias would affect the trained model? Is there a systematic way to mitigate it?

---

> ### Author Response · Authors · 2023-11-14
>
> We thank the Reviewer for the positive feedback.
>
> Questions 1:
>
> We agree with the Referee that the two temperature values of the bi-tempered loss can be considered as two parameters. However, in our case, we did not actively fine-tune them on each dataset but chose two values that were proposed in the original reference, which is now explicitly mentioned in the updated version. We also summarize the parameters in Table 5 in the appendix.
>
>
> Question 2:
>
> This is a great question and we indeed believe that a dynamically tuned $\epsilon$ could further improve performance. We speculate that after the initial training phase, the network gains confidence in a specific label for most examples, i.e., it has a high logit value for one of the classes.
>
> If the largest logit points at the correct class (according to the label) the learning of the example might stop too early if $\epsilon$ is chosen too large, as the network trains to a lower level of confidence (observed for 1000 classes without decreasing $\epsilon$).
> On the other hand, if the largest logit does not correspond to the correct label, which may be due to the label being wrong due to noise,
> the logit bias leads to an increase in the otherwise small logit of the "correct" class. This increases the chances that the network relearns the examples and switches to the "correct" class. In the case of a mislabeled example, this might then reduce the robustness towards label noise, and again is an argument why a dynamical reduction of $\epsilon$ might be useful.
>
> We hence believe that the logit bias has its main use in the early stages of learning. In the later stages, learning is either possible without it as the correct label already gained in confidence, or should not be possible as the example has most likely an incorrect label. Therefore the referees suggestion of a decaying $\epsilon$ schedule could be a great addition to our method.
> We added a brief discussion of this question to the appendix of our manuscript.
>
>
> Question 3:
>
> We do not believe that an initialization scheme which is agnostic of the correct labels can mimic the effects of logit bias, at least not when using softmax as an output function: after initialization, the logits will always have a comparable magnitude, implying that the average magnitude of the softmax output is the inverse of the number of classes. According to our Fig. 1b, this inevitably reduces the overlap between output and the region where the error is large.
>
>
> Question 4:
>
> This question relates to question 2) about the learning behavior after the initial stage has ended. We observed that after the initial phase, most of the examples already have a high enough logit in the correct neuron to allow for learning even with a smaller logit bias.
> In light of this observation, a schedule for the value $\epsilon$ could be useful, such that $\epsilon$ decreases over time.
> This clearly is an interesting question for future research. We have not included this in the present manuscript, since this approach could be considered to be fine-tuning of $\epsilon$, such that to ensure a fair comparison with other loss functions we would have to fine-tune the respective parameters as well, substantially increasing the compute time for numerical experiments.
>
> The problem of having to actively calculate the $\epsilon$ value could principally be mitigated by designing a different loss function that uses our insight into the early stages of learning. By choosing the derivative of the loss function to extend to the middle of the initial distribution independent of the class number we could most likely omit any schedule. However, this is a topic for future research.
>
> We have added a brief discussion of this question to the appendix.

---

### Meta-Review · Area_Chair_5gQm · 2023-12-05

**Metareview:**

This paper analyzed the dynamics of early-stage learning by computing the average backpropagation error, providing quantitative insights into how increasing the class count influences initial learning, especially in the context of bounded loss functions such as Mean Absolute Error (MAE). They introduced a hyperparameter-independent approach called "logit bias," which realigns the distribution of a newly initialized network, enabling effective learning with MAE loss even in scenarios with a multitude of classes. Empirical evidence demonstrates the effectiveness of this method, with logit bias enhanced MAE loss showing comparable or superior performance across datasets spanning ten to a thousand classes. This is significant as such outcomes were previously largely exclusive to Cross Entropy or biTemp loss, which tend to overfit. The authors argue that their method is a first step towards a comprehensive framework that allows for noise-robust learning, regardless of the number of classes, and without an over-reliance on fine-tuned hyperparameters. Specifically, the strength of this paper includes several aspects. 1) This paper offers a novel insight into the underfitting phenomenon observed in some robust loss functions, pinpointing the discord between the non-zero range of the average error and the logit distribution of a newly initialized network as the primary culprit. 2) The approach of this paper is simplicity and efficiency, requiring only a single parameter, which is directly determined by the number of classes. 3) The paper provides clear details about the experimental setup, allowing for reproducibility and further exploration by other researchers.

However, there are several points to be further improved. For example, it is widely known that bounded loss such as MAE suffers from underfitting, an early work also performed gradient analysis to explain this phenomenon. The current experimental results are not sufficient to demonstrate the effectiveness of the proposed method. The proposed method does not achieve the SOTA accuracy. Moreover, this method is only designed for MAE, it should be further expanded to help other robust losses. Therefore, this paper cannot be accepted at ICLR this time, but the enhanced version is highly encouraged to submit other top-tier venues.

**Justification For Why Not Higher Score:**

However, there are several points to be further improved. For example, it is widely known that bounded loss such as MAE suffers from underfitting, an early work also performed gradient analysis to explain this phenomenon. The current experimental results are not sufficient to demonstrate the effectiveness of the proposed method. The proposed method does not achieve the SOTA accuracy. Moreover, this method is only designed for MAE, it should be further expanded to help other robust losses. Therefore, this paper cannot be accepted at ICLR this time, but the enhanced version is highly encouraged to submit other top-tier venues.

**Justification For Why Not Lower Score:**

However, there are several points to be further improved. For example, it is widely known that bounded loss such as MAE suffers from underfitting, an early work also performed gradient analysis to explain this phenomenon. The current experimental results are not sufficient to demonstrate the effectiveness of the proposed method. The proposed method does not achieve the SOTA accuracy. Moreover, this method is only designed for MAE, it should be further expanded to help other robust losses. Therefore, this paper cannot be accepted at ICLR this time, but the enhanced version is highly encouraged to submit other top-tier venues.

---

### Decision · Program_Chairs · 2024-01-16

Reject